# Same-model and cross-model variability in knee cartilage thickness measurements using 3D MRI systems

Hisako Katano[1], Haruka Kaneko[2], Eiji Sasaki[3], Naofumi Hashiguchi[4], Kanto Nagai[5], Muneaki Ishijima[2], Yasuyuki Ishibashi[3], Nobuo Adachi[4], Ryosuke Kuroda[5], Makoto Tomita[6], Jun Masumoto[7], Ichiro Sekiya[1]*

1 Center for Stem Cell and Regenerative Medicine, Institute of Science Tokyo, Tokyo, Japan, 2 Department of Medicine for Orthopedics and Motor Organ, Juntendo University Graduate School of Medicine, Tokyo, Japan, 3 Department of Orthopaedic Surgery, Hirosaki University Graduate School of Medicine, Hirosaki, Aomori, Japan, 4 Department of Orthopaedic Surgery, Graduate School of Biomedical and Health Sciences, Hiroshima University, Hiroshima, Japan, 5 Department of Orthopaedic Surgery, Kobe University Graduate School of Medicine, Kobe, Hyogo, Japan, 6 School of Data Science, Graduate School of Data Science, Yokohama City University, Yokohama, Japan, 7 Medical System Research & Development Center, Fujifilm Corporation, Tokyo, Japan

* sekiya.arm@tmd.ac.jp

## Abstract

### Purpose

Magnetic Resonance Imaging (MRI) based three-dimensional analysis of knee cartilage has evolved to become fully automatic. However, when implementing these measurements across multiple clinical centers, scanner variability becomes a critical consideration. Our purposes were to quantify and compare same-model variability (between repeated scans on the same MRI system) and cross-model variability (across different MRI systems) in knee cartilage thickness measurements using MRI scanners from five manufacturers, as analyzed with a specific 3D volume analysis software.

### Methods

Ten healthy volunteers (eight males and two females, aged 22–60 years) underwent two scans of their right knee on 3T MRI systems from five manufacturers (Canon, Fujifilm, GE, Philips, and Siemens). The imaging protocol included fat-suppressed spoiled gradient echo and proton density weighted sequences. Cartilage regions were automatically segmented into 7 subregions using a specific deep learning-based 3D volume analysis software. This resulted in 350 measurements for same-model variability and 2,800 measurements for cross-model variability.

**Data availability statement:** All relevant data are within the manuscript and its supporting information files.

**Funding:** This study was funded by Fujifilm Corporation.

**Competing interests:** I declare that the authors have no competing interest or other interests that might be perceived to influence the results and/or discussion reported in this paper.

**Abbreviations:**, MRI, magnetic resonance imaging, 3D MRI, three-dimensional MRI, 2D MRI, two-dimensional MRI, OA, osteoarthritis, SPGR, spoiled gradient, PDW, proton-density weighted, DICOM, digital imaging and communications in medicine.

## Results

For same-model variability, 82% of measurements showed variability ≤0.10 mm, and 98% showed variability ≤0.20 mm. For cross-model variability, 51% showed variability ≤0.10 mm, and 84% showed variability ≤0.20 mm. The mean same-model variability (0.06 ± 0.05 mm) was significantly lower than cross-model variability (0.11 ± 0.09 mm) ($p < 0.001$).

## Conclusion

This study demonstrates that knee cartilage thickness measurements exhibit significantly higher variability across different MRI systems compared to repeated measurements on the same system, when analyzed using this specific software. This finding has important implications for multi-center studies and longitudinal assessments using different MRI systems and highlights the software-dependent nature of such variability assessments.

## Introduction

Fully automated three-dimensional analysis of knee MRI, developed through recent advances in medical imaging technology, has enabled the measurement of average cartilage thickness in specific regions of interest in the knee [1–3]. This automated approach offers significant advantages in efficiency and reproducibility compared to manual measurements, potentially improving both clinical practice and research outcomes. Accurate measurement of knee cartilage thickness is crucial for diagnosing and monitoring knee osteoarthritis [4], and this new technology represents a significant advancement in obtaining this measurement [5].

Cartilage thickness measurements can be obtained using any of the currently available MRI systems, all of which offer flexibility in clinical and research settings. Nevertheless, each system introduces its own variability in the measurements, which can affect consistency both between different scanners and within repeated scans on the same system, potentially impacting diagnostic accuracy and the reliability of longitudinal studies. A previous study demonstrated that cartilage thickness could be measured consistently across MRI models from five different manufacturers [6]. However, an unresolved question is whether these measurements maintain the same level of consistency between different models as is observed between repeated scans on the same model.

The purposes of the present study were to evaluate the variability in knee cartilage thickness measurements across MRI systems and to assess data compatibility. We used 3T MRI scanners from five major manufacturers to quantify and compare same-model variability (within the same MRI system through repeated scans) and cross-model variability (between different MRI systems). The findings were expected to contribute to the standardization and improved reliability of knee cartilage thickness measurements by providing crucial insights into the variability between MRI systems and a more accurate comparison of data across different platforms.

## Materials and methods

### Subjects

This study was approved by our medical research ethics committee and written informed consent was obtained from all subjects. The subjects were 10 healthy volunteers aged 22–60 years without any knee complaints. Eight subjects were male, and two subjects were female. Knee radiographs and MRI scans were performed between June 10, 2022, and May 29, 2023. Measurements were made on the right knee of each subject.

### MRI scanning

All 10 right knees were subjected to 3T MRIs using equipment from five different manufacturers (in alphabetical order): a Centurian by Canon in Kobe, Japan; a Trillium Oval by Fujifilm in Tokyo, Japan; an Ingenia by Philips Healthcare in Hiroshima, Japan; a Skyra by Siemens in Tokyo, Japan; and a Signa Premier by GE Medical Systems in Hirosaki, Japan. For each knee joint, images were acquired in the sagittal plane using two different imaging techniques: a fat-suppressed spoiled gradient echo sequence (SPGR) and a proton density-weighted (PDW) sequence (Fig 1a). Prior to scanning the 10 test knees, the right knee of an additional subject was scanned as a preliminary test. This was done to confirm the suitability of the MRI sequences for detecting articular cartilage and to determine the optimal sequence settings for each instrument (Table 1). The sequence parameters were adjusted until clear visualization was achieved. For SPGR sequences, optimization focused on obtaining distinct cartilage contours, while for PDW sequences, the focus was on achieving clear bone contour definition. After the first MRI scan, the subjects stood up once and then underwent a second MRI scan [6,7].

### Automatic segmentation

MRI analyses using deep learning–based segmentation were performed using a 3D volume analysis software (SYNAPSE 3D [Japanese product name: SYNAPSE VINCENT], Collaborative version 6.7, Fujifilm Corporation, Tokyo, Japan). The automatic segmentation algorithm was trained on SPGR and PDW images from 101 datasets obtained from Philips, nine from Canon, five from GE, two from Fujifilm, and none from Siemens. The bone region was automatically segmented from the PDW images, and the cartilage region was automatically segmented from the SPGR images. The 3D image was then reconstructed (Fig 1b) [8,9].

### Measurements of cartilage thickness

The cartilage region of interest (ROI) was derived by creating 3D images of the bone, followed by definition based on the bone's shape and the smoothness of its surface. Regions located 2 mm inside this defined cartilage area were set as the regions of interest (ROIs) for measuring cartilage thickness (Fig 1b).

The femoral cartilage was projected along the long axis of the femur. The rotation was determined by generating a horizontal tangent line between the ROI at the posteromedial cartilage and the ROI at the posterolateral cartilage. The software drew lines that split the ROI equally in the longitudinal and transverse directions, resulting in 4 regions. Of these regions, the anterior medial femoral (AMF) and anterior lateral femoral (ALF) regions represent non-load-bearing areas, while the posterior medial femoral (PMF) and posterior lateral femoral (PLF) regions serve as load-bearing areas of the knee joint. The tibial cartilage was also projected along the long axis of the tibia, and the ROIs for the medial and lateral tibias were automatically drawn as two closed curve lines. The patellar cartilage was projected so that the area reached its maximum, and the ROI was automatically drawn as a closed curve line (Fig 1b) [10].

A visual map of the cartilage thickness was provided using a color scale, with thicker areas indicated in white and thinner areas indicated in red. The software segmented the cartilage overlying the most superficial layer of the bone into the smallest detectable units, quantified the thickness for each unit, and provided the average cartilage thickness of the ROI [11–13].

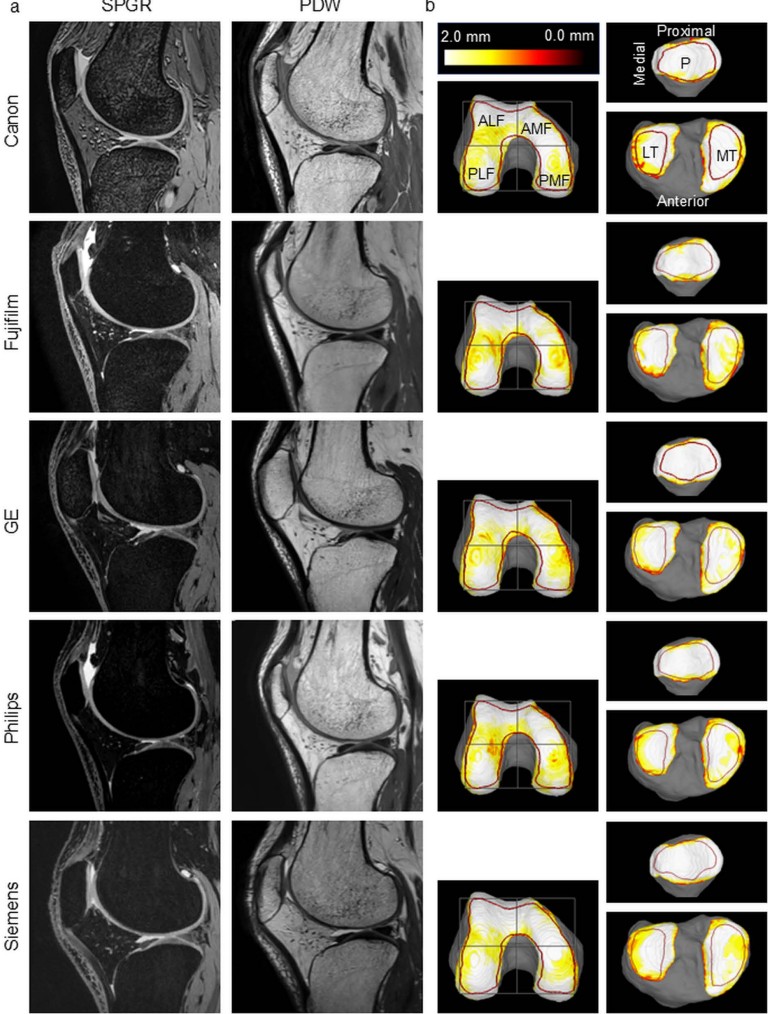

**Fig 1. MR images of the knee and cartilage thickness mapping obtained using MRI systems from different manufacturers.** (a) Sagittal MR images of the knee taken with MRI machines from five different companies. Representative examples were selected from each company. (b) MRI cartilage thickness mapping and regions of the knee. The cartilage thickness was provided using a color scale, with thicker areas indicated in white and thinner areas indicated in red. The region of interest is indicated by a red line, and the femoral cartilage is divided into four regions by halving it vertically and horizontally. PMF, posterior medial femoral; PLF, posterior lateral femoral; AMF, anterior medial femoral; ALF, anterior lateral femoral; MT, medial tibial; LT, lateral tibial; P, patellar.

The cartilage thicknesses obtained by evaluating the MR images from the five models were plotted for seven regions across the 10 subjects. The cartilage thickness variations produced by the MRI models were also analyzed in the same seven regions.

## Same-model variability and cross-model variability

The absolute difference in cartilage thickness between the first and second scans on the same MRI model was defined as same-model variability. This resulted in five combinations for one region of one knee (Fig 2). MRI scans were performed on the 10 subjects using the five different MRI models, and measurements were made for the seven knee regions. The data were plotted separately for the first and second scans; thus, the same-model variability was represented by a total of 350 plots.

**Table 1. Imaging parameters for the MRI sequences.**

| Company | Canon | Fujifilm | GE | Philips | Siemens |
|---|---|---|---|---|---|
| Model Name | Centurian | TRILLIUM_OVAL | SIGNA Premier | Ingenia | Skyra |
| Receive Coil Name | 16chFlex SPDR M | Extremity | 18Knee | SENSE KNEE 16 AC | TxRx Knee 15 |
| Transmit Coil Name | QD Whole Body | TR Body | 18Knee | BODY | TxRx Knee 15 |
| **SPGR** | | | | | |
| Slice Thickness (mm) | 0.70 | 0.80 | 0.60 | 0.60 | 0.36 |
| Repetition Time (ms) | 20.80 | 8.40 | 11.30 | 23.19 | 13.46 |
| Echo Time (ms) | 11.40 | 4.20 | 4.90 | 6.91 | 5.00 |
| Echo Number(s) | 1.00 | 1.00 | 1.00 | 1.00 | 1.00 |
| Magnetic Field Strength (T) | 3.00 | 2.90 | 3.00 | 3.00 | 3.00 |
| Spacing Between Slices (mm) | 0.35 | 0.40 | 0.30 | 0.30 | 0.00 |
| Number of Phase-Encoding Steps | 136 | 256 | Not calculated | 254 | 223 |
| Echo Train Length | 1 | 1 | 1 | 3 | 2 |
| Percent Phase Field of View (%) | 100 | 100 | 100 | 100 | 100 |
| Pixel Bandwidth (Hz) | 279 | 547 | 98 | 310 | 285 |
| Acquisition Matrix* | 0/272/272/0 | 0/256/256/0 | 0/320/320/0 | 0/256/254/0 | 0/256/256/0 |
| Pixel Spacing (mm)** | 0.29/0.29 | 0.31/0.31 | 0.31/0.31 | 0.29/0.29 | 0.29/0.29 |
| Flip Angle (degree) | 4 | 20 | 25 | 30 | 15 |
| SAR (w/kg) | 0.56 | 0.13 | 0.03 | 0.01 | 0.01 |
| dB/dt (T/s) | 8.59 | 0.67 | Not calculated | 96.33 | 0.00 |
| **PDW** | | | | | |
| Slice Thickness (mm) | 0.70 | 0.80 | 0.59 | 0.60 | 0.36 |
| Repetition Time (ms) | 1100 | 1500 | 1000 | 1000 | 1000 |
| Echo Time (ms) | 13.00 | 34.40 | 31.65 | 31.47 | 34.00 |
| Echo Number(s) | 1.00 | 1.00 | 1.00 | 1.00 | 1.00 |
| Magnetic Field Strength (T) | 3.00 | 2.90 | 3.00 | 3.00 | 3.00 |
| Spacing Between Slices (mm) | 0.35 | 0.40 | 0.30 | 0.30 | 0.00 |
| Number of Phase-Encoding Steps | 80 | 224 | Not calculated | 254 | 123 |
| Echo Train Length | 18 | 38 | 60 | 40 | 24 |
| Percent Phase Field of View (%) | 100 | 100 | 100 | 100 | 100 |
| Pixel Bandwidth (Hz) | 488 | 625 | 244 | 592 | 330 |
| Acquisition Matrix* | 0/320/320/0 | 0/224/224/0 | 0/320/320/0 | 0/256/254/0 | 0/320/208/0 |
| Pixel Spacing (mm)** | 0.25/0.25 | 0.33/0.33 | 0.31/0.31 | 0.29/0.29 | 0.23/0.23 |
| Flip Angle (degree) | 90 | 90 | 90 | 90 | 120 |
| SAR (w/kg) | 0.64 | 0.65 | 0.26 | 0.04 | 0.05 |
| dB/dt (T/s) | 13.96 | 0.72 | Not calculated | 87.31 | 0.00 |

SPGR = fat-suppressed spoiled gradient echo; PDW = proton density weighted; SAR = specific absorption rate; dB/dt = rate of change of the magnetic field. * = Frequency Row/ Frequency Column/ Phase Row/ Phase Column, ** = in the phase encoding/ read direction.

The absolute difference between measurements made using one MRI model and those made using the other models was defined as the cross-model variability. This resulted in 40 combinations for one region of one knee (Fig 2). The 40 combinations for each region and subject resulted in 400 plots per region. Thus, the seven regions gave a total of 2,800 plots representing cross-model variability.

Histograms were generated to show the same-model variability (350 measurements) and the cross-model variability (2,800 measurements) of the cartilage thickness measurements. The proportions of all measurements with a variability of 0.10 mm or less and of 0.20 mm or less were determined.

| | Canon 1st | Canon 2nd | Fujifilm 1st | Fujifilm 2nd | GE 1st | GE 2nd | Philips 1st | Philips 2nd | Siemens 1st | Siemens 2nd |
|---|---|---|---|---|---|---|---|---|---|---|
| Canon 1st | Baseline | | | | | | | | | |
| Canon 2nd | Same | Baseline | | | | | | | | |
| Fujifilm 1st | Cross | Cross | Baseline | | | | | | | |
| Fujifilm 2nd | Cross | Cross | Same | Baseline | | | | | | |
| GE 1st | Cross | Cross | Cross | Cross | Baseline | | | | | |
| GE 2nd | Cross | Cross | Cross | Cross | Same | Baseline | | | | |
| Philips 1st | Cross | Cross | Cross | Cross | Cross | Cross | Baseline | | | |
| Philips 2nd | Cross | Cross | Cross | Cross | Cross | Cross | Same | Baseline | | |
| Siemens 1st | Cross | Cross | Cross | Cross | Cross | Cross | Cross | Cross | Baseline | |
| Siemens 2nd | Cross | Cross | Cross | Cross | Cross | Cross | Cross | Cross | Same | Baseline |

Same, same-model variability
Cross, cross-model variability

**Fig 2. Combination of same-model variability and cross-model variability.** For one knee, MRI scans were taken twice using the same MRI model. These scans were repeated for the five different MRI models. The absolute difference in cartilage thickness between the first and second scans on the same model was defined as same-model variability. This resulted in five combinations for one region of one knee. The absolute difference between measurements on one model and measurements on other models was defined as the cross-model variability. This resulted in 40 combinations for one region of one knee.

## Statistical analysis

Graphs of individual cartilage thickness and variability, plotted by subject and by region in corporate colors, were generated using GraphPad Prism 10 (GraphPad Software, Boston, MA, USA). Comparisons between same-model variability and cross-model variability were tested using Welch's t-test with a significance level set at 0.05. The BellCurve software for Excel (Social Survey Research Information Co., Ltd., Tokyo, Japan) was used for this analysis. Histograms of the inter-measurement error in cartilage thickness were prepared using Excel 2021 (Microsoft, Redmond, WA, USA).

## Results

### Cartilage thickness

Cartilage thickness was measured at seven regions in the 10 subjects using MRI models from five companies. The measurements were plotted for the first and second time points (Fig 3). In both scatter plots, although differences were observed among subjects and companies, the P (patellar) region consistently showed the greatest thickness, followed by the LT (lateral tibial) region, when comparing the maximum and minimum values for each region.

### Scatter plots of same-model variability and cross-model variability

None of the same-model variability measurements exceeded 0.4 mm, while some cross-model variability measurements surpassed 0.5 mm (Fig 4). For same-model variability, the largest values were observed in two measurements from the P (patellar) region, followed by one measurement from the LT (lateral tibial) region. Regarding cross-model variability, the largest measurement was found in one measurement from the MT (medial tibial) region, followed by one measurement each from the MT, LT, and P regions.

### Distribution of same-model variability and cross-model variability

For the same-model variability, 82% of the 350 measurements showed variability ≤ 0.10 mm and 98% showed variability ≤ 0.20 mm (Fig 5). For the cross-model variability, 51% of the 2,800 measurements showed variability ≤ 0.10 mm and

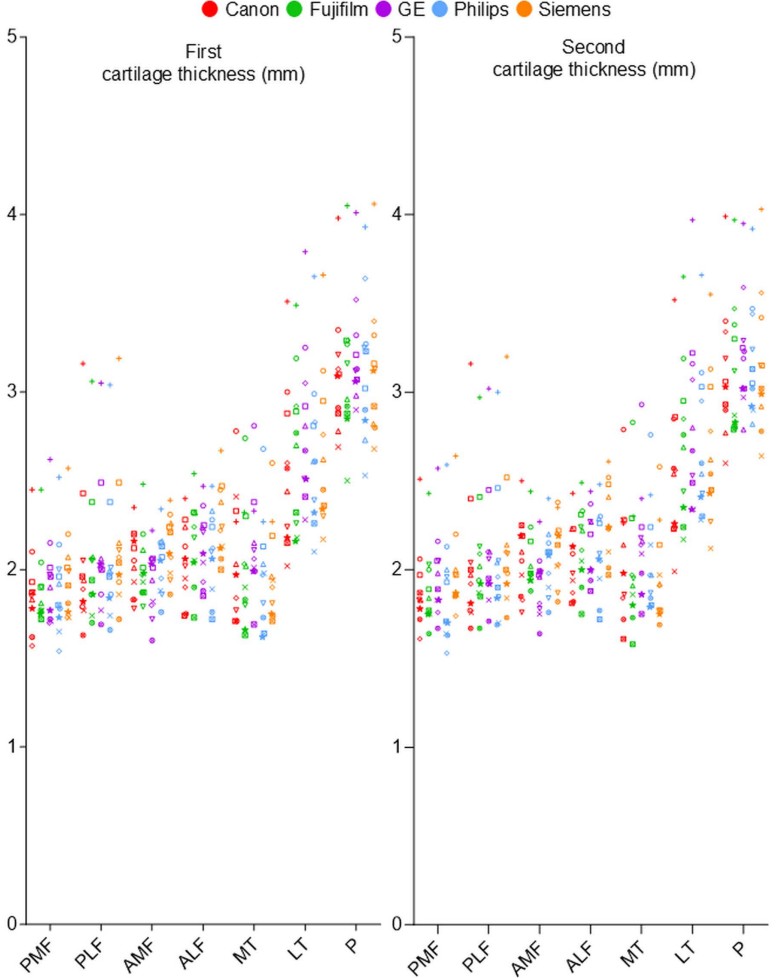

**Fig 3. Cartilage thickness measured at the first and second time points.** MRI scans were performed on 10 subjects using five different MRI models. Each subject was represented by a standardized symbol. Each MRI model was displayed in its own corporate color. Measurements were shown for seven knee regions. The data were plotted separately for the first and second scans. This yielded a total of 350 plots.

84% showed variability ≤ 0.20 mm. The mean ± SD was 0.06 ± 0.05 mm for same-model variability and 0.11 ± 0.09 mm for cross-model variability. Welch's t-test revealed that cross-model variability was significantly higher than same-model variability ($p < 0.001$).

## Discussion

This study investigated the variability in knee cartilage thickness measurements determined using fully automatic three-dimensional analysis of scans from MRI systems from five different manufacturers. We compared same-model variability (between repeated scans on the same MRI system) and cross-model variability (across different MRI systems) in 10 healthy volunteers. Our results showed that cross-model variability was significantly higher than same-model variability. While measurements from identical MRI systems demonstrated high consistency with deviations of 0.10 mm or less in the vast majority of cases, only 67% of the cross-model measurements achieved this level of precision. The mean same-model variability (0.06 ± 0.05 mm) was significantly lower than the mean cross-model variability (0.12 ± 0.09 mm). These

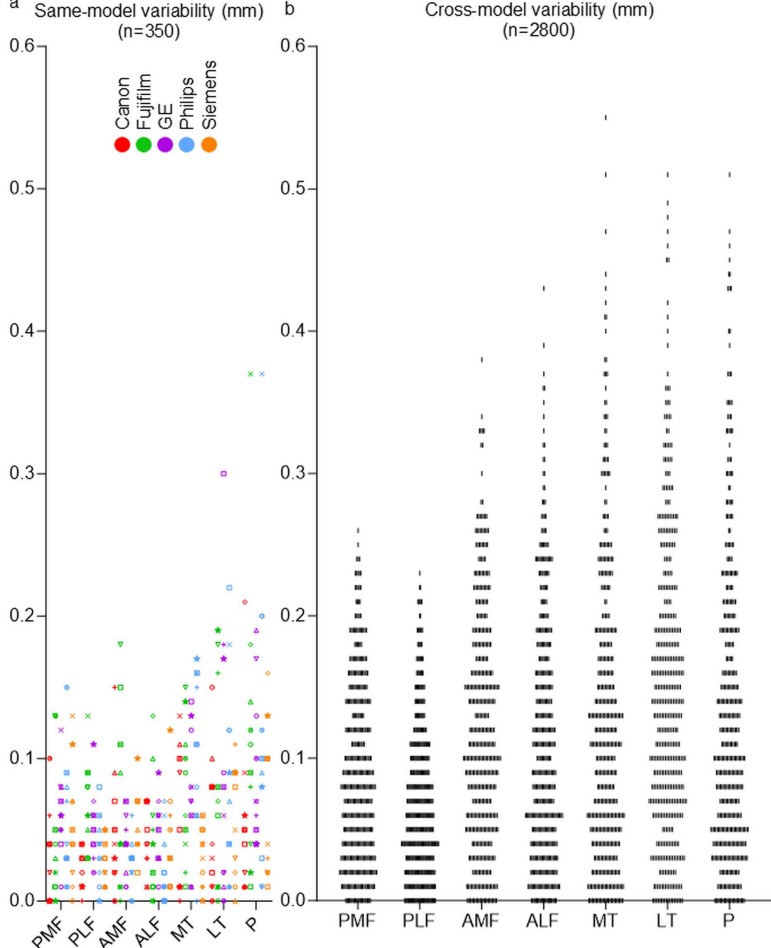

**Fig 4. Scatter plots of same-model variability and cross-model variability.** (a) Scatter plot of same-model variability. Each subject was represented by a standardized symbol. Each MRI model was displayed in its own corporate color. Measurements are shown for seven regions. There were five combinations for each region and subject, resulting in 50 plots per region. The seven knee regions gave a total of 350 plots. (b) Scatter plot of cross-model variability. For each region, the cross-model variability was plotted as a single vertical bar, without distinguishing among the 10 subjects and five MRI models. This yielded 40 combinations for each region and subject, resulting in 400 plots per region. The seven knee regions gave a total of 2,800 plots.

findings highlight the importance of considering MRI system variability when comparing cartilage thickness measurements across different scanners or in longitudinal studies.

Cartilage thickness measurements showed general consistency across subjects in the 3D analysis of 10 knees scanned using MRI systems from five different manufacturers. However, when the same 10 knees were scanned twice, the cross-model variability exceeded the same-model variability. The primary sources of this variation can be attributed to two main factors: differences in MRI images as sequence parameters such as slice thickness could be reduced by standardizing these parameters [14,15], and variations in automatic articular cartilage extraction and ROI settings which might yield different results with alternative segmentation software. The difference in sample sizes has likely contributed to the observed variations, with 350 measurements for same-model versus 2,800 for cross-model comparisons. Larger sample sizes tend to capture a wider range of measurement fluctuations, potentially amplifying the differences between MRI models [16]. These findings, based on specific MRI systems and software, require further validation for applicability to other

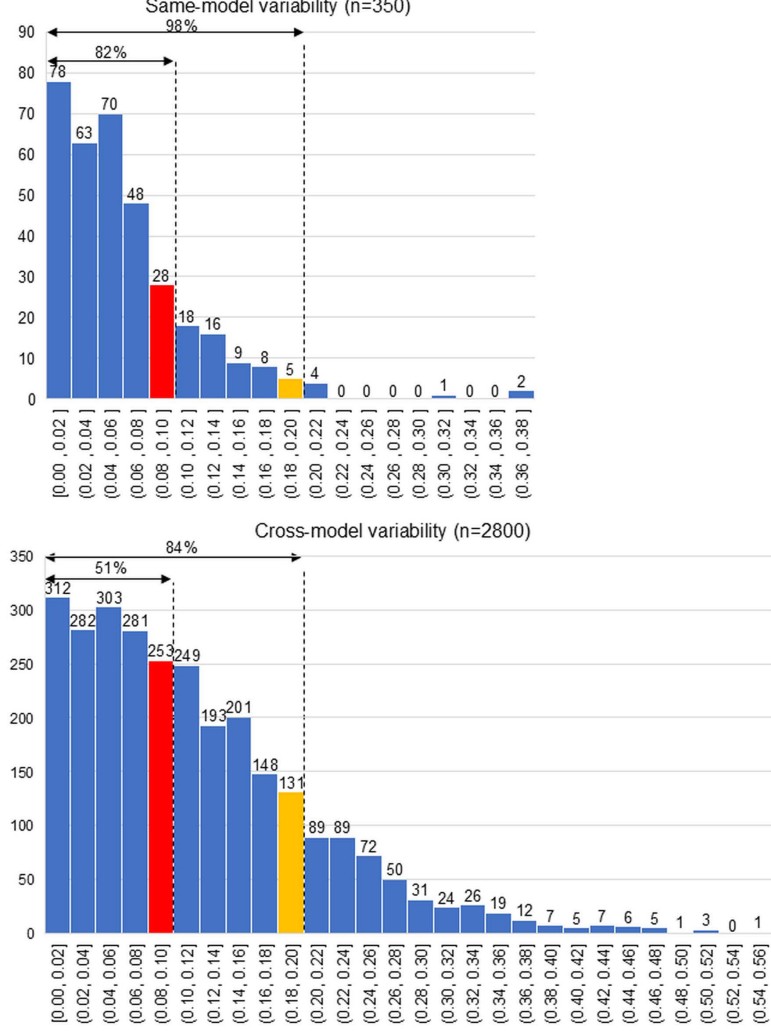

**Fig 5. Distribution of same-model variability and cross-model variability.** The number of measurements is shown above each bar in the graph. For example, a variability of "(0.08, 0.10) mm" indicates 0.08 mm < variability ≤ 0.10 mm. For the same-model variability, the variability was ≤ 0.10 mm in 84% of the 350 measurements and ≤ 0.20 mm in 98%. For the cross-model variability, the variability was ≤ 0.10 mm in 67% of the 2800 measurements and ≤ 0.20 mm in 83%.

scanner models, vendors, and different field strengths, suggesting that broader verification would be beneficial before recommending specific software for cartilage thickness measurement studies.

The same-model variability in the current study was 0.06 ± 0.05 mm, with 84% of the measurements showing variability ≤ 0.10 mm and 98% ≤ 0.20 mm. While this finding deviates from the study's main objective, it suggests that for subjects without knee symptoms, a change of 0.20 mm in cartilage thickness following an intervention has a 98% probability of being significant. Eckstein et al., using semi-automatic 3D MRI analysis of knees with worsening pain (n = 194), demonstrated a decrease of 0.18 mm in central medial femur cartilage thickness over two years [17]. Our results support the interpretation that this type of change likely represents a true biological change rather than a measurement error. Consequently, the same-model variability established in this study can serve as a reliable benchmark for evaluating cartilage thickness changes in longitudinal and interventional studies.

The threshold of 0.10 mm used in this study was selected as a reasonable intermediate value between 1.00 mm, which would be too coarse for detecting subtle cartilage changes, and 0.01 mm, which would be unrealistically precise given current MRI resolution capabilities. Previous studies have reported that annual cartilage thinning in osteoarthritis typically ranges from 0.15 to 0.30 mm [18], with rates varying by region and disease severity [19,20]. Early-stage osteoarthritis can show even more subtle changes, making precise measurement crucial for early detection. Therefore, measurement variability should ideally be lower than these expected pathological changes to ensure reliable detection of disease progression. In our study, same-model variability achieved this criterion with 82% of measurements showing variability ≤0.10 mm, suggesting adequate precision for longitudinal monitoring when using the same scanner. However, cross-model variability met this threshold in only 51% of measurements, indicating potential challenges in cross-scanner comparisons. These findings suggest that while 0.10 mm represents a clinically meaningful and technically reasonable threshold for cartilage thickness measurements, achieving this level of precision consistently across different manufacturers' systems remains challenging with current technology.

The training data for our automatic segmentation algorithm had an uneven distribution across manufacturers. Despite this imbalance in the training data, our previous study demonstrated that cartilage thickness measurements were consistent across different vendors when examining 10 healthy subjects using MRI scanners from five manufacturers [6]. However, previous studies using deep learning for medical image analysis have demonstrated that the diversity and balance of training data can affect model performance and generalizability across MRI scanners from different manufacturers [21–23]. To optimize the algorithm's performance and ensure consistent accuracy across all manufacturers, future work should include a more balanced distribution of training data from different MRI vendors. This is particularly important as image acquisition parameters and scanner characteristics could potentially affect segmentation accuracy.

The distribution of measurement variability showed distinct patterns between same-model and cross-model comparisons. We chose Welch's t-test over Student's t-test because the two groups had unequal sample sizes (n = 350 for same-model and n = 2,800 for cross-model measurements) and unequal variances (SD = 0.05 mm and 0.09 mm, respectively). The F-test for equality of variances confirmed significantly different variances between the groups (p < 0.001). In such cases, Welch's t-test is more robust than Student's t-test as it does not assume equal variances [24], making it the more appropriate choice for our analysis.

A few limitations should be considered when interpreting the results of this study. First, it is important to emphasize that our findings are specific to the particular 3D volume analysis software (SYNAPSE 3D) that was used in this study. Different segmentation algorithms or analysis software might yield different variability patterns across manufacturers. The software-specific nature of our results limits their direct applicability to studies using alternative analysis platforms. Second, the sample size was limited to only 10 subjects, which may restrict the generalizability of our findings. Third, our study focused solely on healthy volunteers without knee complaints. A more diverse cohort with a balanced male-to-female ratio and including subjects of various ages and with mild to severe OA could provide more comprehensive and clinically relevant results. The variability in cartilage thickness measurements might differ in patients with osteoarthritis or other knee conditions due to altered cartilage states. Finally, our assessment was limited to two scans per subject, providing only a short-term evaluation. A longitudinal study with multiple time points could offer insights into long-term variability and the stability of measurements over time, potentially revealing how variability across MRI systems might change with disease progression.

In conclusion, this study shows that cartilage measurements vary more between different MRI models than within the same model. This higher variation across models means that comparisons of data from different MRI systems or from studies involving multiple centers should be evaluated with caution. Our findings help improve the reliability and standardization of knee cartilage thickness measurements for both clinical use and research.

## Supporting information

**S1 Table. Dataset.**
(XLSX)

## Acknowledgments

We thank Ms. Sayaka Komura and Ms. Chiaki Okumura for managing our laboratory. This manuscript was proofread by Ms. Ellen Roider.

## Author contributions

**Conceptualization:** Ichiro Sekiya.

**Data curation:** Hisako Katano, Ichiro Sekiya.

**Formal analysis:** Hisako Katano, Makoto Tomita, Ichiro Sekiya.

**Funding acquisition:** Ichiro Sekiya.

**Investigation:** Hisako Katano, Haruka Kaneko, Eiji Sasaki, Naofumi Hashiguchi, Kanto Nagai, Muneaki Ishijima, Yasuyuki Ishibashi, Nobuo Adachi, Ryosuke Kuroda, Ichiro Sekiya.

**Methodology:** Ichiro Sekiya.

**Project administration:** Ichiro Sekiya.

**Resources:** Hisako Katano, Ichiro Sekiya.

**Software:** Jun Masumoto.

**Supervision:** Ichiro Sekiya.

**Validation:** Hisako Katano, Ichiro Sekiya.

**Visualization:** Hisako Katano, Ichiro Sekiya.

**Writing – original draft:** Ichiro Sekiya.

**Writing – review & editing:** Hisako Katano, Ichiro Sekiya.

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
