## [Decision Letter · Decision Letter 0]

25 Feb 2025

Dear Dr. Sekiya,

Thank you for submitting your manuscript to PLOS ONE. After careful consideration, we feel that it has merit but does not fully meet PLOS ONE’s publication criteria as it currently stands. Therefore, we invite you to submit a revised version of the manuscript that addresses the points raised during the review process.

We look forward to receiving your revised manuscript.

Kind regards,

Ewa Tomaszewska, DVM Ph.D

Academic Editor

PLOS ONE

3. In the online submission form, you indicated that [The data used in this study are available from the corresponding author upon reasonable request.].

Additional Editor Comments (if provided):

Reviewers' comments:

Reviewer's Responses to Questions

**Comments to the Author**

1. Is the manuscript technically sound, and do the data support the conclusions?

Reviewer #1: Yes

2. Has the statistical analysis been performed appropriately and rigorously?

Reviewer #1: Yes

3. Have the authors made all data underlying the findings in their manuscript fully available?

Reviewer #1: No

4. Is the manuscript presented in an intelligible fashion and written in standard English?

Reviewer #1: Yes

Reviewer #1: The authors of this paper studied same-model and cross-model variability in knee cartilage thickness measurements on 3D images obtained by five manufacturer MRI scanners.

It is a niche study that, anyhow, allows to have reference values for intra and inter scanner uncertainties. Sometimes the results are weak: specifically it is obvious that cross-model variability was higher than same-model variability.

The text is clear and really well written.

The statistical approach is correct.

I believe that the title is overfull and should have less information: the number of manufacturers for example is not primary in a title.

In the text, including the abstract and the limitations in the Discussion section, should be more stressed that the results are specific for the "3D volume analysis software" that has been used.

Minor concerns:

1) in the "Automatic Segmentation" subsection, second line: "using A 3D volume...";

2) in the "Automatic Segmentation" subsection, fifth line: "from 101 datasets...";

3) in the "Measurements of Cartilage Thickness" subsection, 18-19 lines should be referenced the Figure 1b.

**Do you want your identity to be public for this peer review?** For information about this choice, including consent withdrawal, please see our Privacy Policy

Reviewer #1: No

---

## [Author Response · Author response to Decision Letter 1]

21 Apr 2025

Reviewer #1: The authors of this paper studied same-model and cross-model variability in knee cartilage thickness measurements on 3D images obtained by five manufacturer MRI scanners.

It is a niche study that, anyhow, allows to have reference values for intra and inter scanner uncertainties. Sometimes the results are weak: specifically it is obvious that cross-model variability was higher than same-model variability.

The text is clear and really well written.

The statistical approach is correct.

We sincerely thank the reviewer for their positive assessment of our manuscript, particularly noting that our text is 'clear and really well written' and that our statistical approach is 'correct.' We acknowledge the reviewer's observation that the finding of higher cross-model variability compared to same-model variability may seem intuitive. However, we believe that quantifying these differences provides essential reference values that have not been previously documented in the literature for knee cartilage measurements using modern 3D MRI technology. These reference values serve as critical benchmarks for multi-center studies and longitudinal assessments that utilize different MRI systems. We appreciate the reviewer's recognition of the utility of our work in establishing these reference values for intra and inter-scanner uncertainties.

I believe that the title is overfull and should have less information: the number of manufacturers for example is not primary in a title.

We have changed from “Five-manufacturer 3D MRI Study on Same-Model and Cross-Model Variability in Knee Cartilage Thickness Measurements” to “Same-Model and Cross-Model Variability in Knee Cartilage Thickness Measurements Using 3D MRI systems”.

In the text, including the abstract and the limitations in the Discussion section, should be more stressed that the results are specific for the "3D volume analysis software" that has been used.

We have modified the abstract as follows:

“Purpose: Magnetic Resonance Imaging (MRI) based three-dimensional analysis of knee cartilage has evolved to become fully automatic. However, when implementing these measurements across multiple clinical centers, scanner variability becomes a critical consideration. Our purposes were to quantify and compare same-model variability (between repeated scans on the same MRI system) and cross-model variability (across different MRI systems) in knee cartilage thickness measurements using MRI scanners from five manufacturers, as analyzed with a specific 3D volume analysis software.

Methods: Ten healthy volunteers (eight males and two females, aged 22-60 years) underwent two scans of their right knee on 3T MRI systems from five manufacturers (Canon, Fujifilm, GE, Philips, and Siemens). The imaging protocol included fat-suppressed spoiled gradient echo and proton density weighted sequences. Cartilage regions were automatically segmented into 7 subregions using a specific deep learning-based 3D volume analysis software. This resulted in 350 measurements for same-model variability and 2,800 measurements for cross-model variability.

Results: For same-model variability, 82% of measurements showed variability ≤0.10 mm, and 98% showed variability ≤0.20 mm. For cross-model variability, 51% showed variability ≤0.10 mm, and 84% showed variability ≤0.20 mm. The mean same-model variability (0.06 ± 0.05 mm) was significantly lower than cross-model variability (0.11 ± 0.09 mm) (p<0.001).

Conclusion: This study demonstrates that knee cartilage thickness measurements exhibit significantly higher variability across different MRI systems compared to repeated measurements on the same system, when analyzed using this specific software. This finding has important implications for multi-center studies and longitudinal assessments using different MRI systems and highlights the software-dependent nature of such variability assessments.”

We have also modified the limitations as follows:

“A few limitations should be considered when interpreting the results of this study. First, it is important to emphasize that our findings are specific to the particular 3D volume analysis software (SYNAPSE 3D) that was used in this study. Different segmentation algorithms or analysis software might yield different variability patterns across manufacturers. The software-specific nature of our results limits their direct applicability to studies using alternative analysis platforms. Second, the sample size was limited to only 10 subjects, which may restrict the generalizability of our findings. Third, our study focused solely on healthy volunteers without knee complaints. A more diverse cohort with a balanced male-to-female ratio and including subjects of various ages and with mild to severe OA could provide more comprehensive and clinically relevant results. The variability in cartilage thickness measurements might differ in patients with osteoarthritis or other knee conditions due to altered cartilage states. Finally, our assessment was limited to two scans per subject, providing only a short-term evaluation. A longitudinal study with multiple time points could offer insights into long-term variability and the stability of measurements over time, potentially revealing how variability across MRI systems might change with disease progression.”

Minor concerns:

1) in the "Automatic Segmentation" subsection, second line: "using A 3D volume...";

We have corrected as follows: “MRI analyses using deep learning–based segmentation were performed using a 3D volume analysis software (SYNAPSE 3D [Japanese product name: SYNAPSE VINCENT], Collaborative version 6.7, Fujifilm Corporation, Tokyo, Japan).”

2) in the "Automatic Segmentation" subsection, fifth line: "from 101 datasets...";

We have corrected as follows: “The automatic segmentation algorithm was trained on SPGR and PDW images from on 101 datasets obtained from Philips, nine from Canon, five from GE, two from Fujifilm, and none from Siemens.”

3) in the "Measurements of Cartilage Thickness" subsection, 18-19 lines should be referenced the Figure 1b.

We have corrected as follows: “The patellar cartilage was projected so that the area reached its maximum, and the ROI was automatically drawn as a closed curve line (Figure 1b) (10).”

---

## [Decision Letter · Decision Letter 1]

4 May 2025

Same-model and cross-model variability in knee cartilage thickness measurements using 3D MRI systems

PONE-D-25-00129R1

Dear Dr. Ichiro Sekiya,

We’re pleased to inform you that your manuscript has been judged scientifically suitable for publication and will be formally accepted for publication once it meets all outstanding technical requirements.

Kind regards,

Ewa Tomaszewska, DVM Ph.D

Academic Editor

PLOS ONE

Additional Editor Comments (optional):

Reviewers' comments:

Reviewer's Responses to Questions

**Comments to the Author**

Reviewer #1: All comments have been addressed

2. Is the manuscript technically sound, and do the data support the conclusions?

Reviewer #1: Yes

3. Has the statistical analysis been performed appropriately and rigorously?

Reviewer #1: Yes

4. Have the authors made all data underlying the findings in their manuscript fully available?

Reviewer #1: Yes

5. Is the manuscript presented in an intelligible fashion and written in standard English?

Reviewer #1: Yes

Reviewer #1: The authors of this paper have answered all my comments and fully addressed my concerns updating the text accordingly.

**Do you want your identity to be public for this peer review?** For information about this choice, including consent withdrawal, please see our Privacy Policy

Reviewer #1: No

---

## [Editor Report · Acceptance letter]

PONE-D-25-00129R1

PLOS ONE

Dear Dr. Sekiya,

I'm pleased to inform you that your manuscript has been deemed suitable for publication in PLOS ONE. Congratulations! Your manuscript is now being handed over to our production team.

Kind regards,

on behalf of

Professor Ewa Tomaszewska

Academic Editor

PLOS ONE